# In Silico and In Vitro Analyses Validate Human MicroRNAs Targeting the SARS-CoV-2 3′-UTR

**DOI:** 10.3390/ijms22116094

**Published:** 2021-06-05

**Authors:** María Asunción Barreda-Manso, Manuel Nieto-Díaz, Altea Soto, Teresa Muñoz-Galdeano, David Reigada, Rodrigo M. Maza

**Affiliations:** Molecular Neuroprotection Group, Research Unit, National Hospital for Paraplegics (SESCAM), 45071 Toledo, Spain; mbarreda@sescam.jccm.es (M.A.B.-M.); alteas@sescam.jccm.es (A.S.); tmunozd@sescam.jccm.es (T.M.-G.); dreigada@sescam.jccm.es (D.R.)

**Keywords:** SARS-CoV-2 3′-UTR, hsa-miR-3941, hsa-miR-138-5p, antiviral defense

## Abstract

COVID-19 pandemic is caused by betacoronavirus SARS-CoV-2. The genome of this virus is composed of a single strand of RNA with 5′ and 3′-UTR flanking a region of protein-coding ORFs closely resembling cells’ mRNAs. MicroRNAs are endogenous post-transcriptional regulators that target mRNA to modulate protein expression and mediate cellular functions, including antiviral defense. In the present study, we carried out a bioinformatics screening to search for endogenous human microRNAs targeting the 3′-UTR of SARS-CoV-2. Results from the computational techniques allowed us to identify 10 potential candidates. The capacity of 3 of them, together with hsa-miR-138-5p, to target the SARS-CoV-2 3′-UTR was validated in vitro by gene reporter assays. Available information indicates that two of these microRNAs, namely, hsa-miR-3941 and hsa-miR-138-5p, combine effective targeting of SARS-CoV-2 genome with complementary antiviral or protective effects in the host cells that make them potential candidates for therapeutic treatment of most, if not all, COVID-19 variants known to date. All information obtained while conducting the present analysis is available at Open Science Framework repository.

## 1. Introduction

Coronavirus disease 2019 (COVID-19) is an infectious disease caused by severe acute respiratory syndrome coronavirus 2 (SARS-CoV-2). Common symptoms include fever, cough, and shortness of breath that can progress to severe pneumonia and multi-organ failure, although the majority of cases result in mild symptoms and some infected people may be asymptomatic. By the end of May 2021, more than 167 million people had become infected worldwide since the beginning of the pandemic and more than 3.4 million had died [1]. Currently, options for the management of COVID-19 patients are still limited to symptom treatment, supportive care, isolation, and experimental treatments. Initial expectations on the repurposed antiviral drugs remdesivir, hydroxychloroquine, lopinavir, and interferon beta-1a proved to be premature as they “had little or no effect on hospitalized patients with COVID-19, as indicated by overall mortality, initiation of ventilation, and duration of hospital stay” according to a worldwide clinical trial organized by the World Health Organization [2].

MicroRNA-based therapies represent an alternative that remains to be fully explored. MicroRNAs are short non-coding RNA sequences that bind to target messenger RNAs (mRNAs), causing their cleavage or translational block to modulate a range of fundamental cellular processes (for an updated description of microRNA biology see [3]). MicroRNA-mediated RNA interference provides a layer of post-transcriptional gene regulation employed by virus and host to enhance or inhibit viral infection [4,5] with potential therapeutic applications currently under clinical trial [6]. Accumulating evidence indicates that host cellular microRNAs can target the 3′-untranslated region (3′-UTR) and coding regions of RNA virus to repress their translation and mediate antiviral defense [7,8,9].

SARS-CoV-2 genome is composed of a positive-sense single-stranded 30 Kb long RNA that structurally resembles eukaryotic mRNAs. The sequence comprises 12 putative coding regions flanked by a 5′-UTR methylated cap and a 3′-UTR with a poly-A tail [10]. Resemblance of SARS-CoV-2 genome to mRNAs suggests that it may contain binding sites for microRNAs as observed in most mammal genes. Indeed, published and pre-printed in silico studies have identified putative binding sites in SARS-CoV-2 genome for human microRNAs [11,12,13,14,15,16,17], for some of which antiviral effects have been shown against influenza A, Ebola, and hepatitis B (HBV) virus infections [15,18]. However, besides the 3′-UTR is acknowledged as the canonical target region for microRNAs and its regulation has the potential to reduce the expression and translation of the whole viral genome, information on the microRNA binding sites in this region is still very fragmentary. Moreover, to our knowledge, no study has confirmed the binding and regulatory capacities of any of the in silico predicted human microRNAs over the SARS-CoV-2 expression. In this study, we have combined in silico analysis with reporter assays in human cell cultures to identify human microRNAs targeting the 3′-UTR of SARS-CoV-2, which may be employed to regulate virus replication and expression. All obtained information is available at Open Science Framework (OSF, https://osf.io/9drpj/) and as the Appendix A of this article.

## 2. Results

### 2.1. Prediction Tools Indicate That Human MicroRNAs Have Binding Sites in the 3′-UTR Sequence of SARS-CoV-2 Genome

Identifying human microRNAs with binding sites in the 3′-UTR of SARS-CoV-2 genome can be achieved through multiple predictive tools that take into account different binding features and experimental data (for review see [19]). Among them, we decided to employ miRDB, IntaRNA, RNA22, RNAhybrid, and STarMir predictive tools because, contrary to others, they do not restrict the analysis to predefined sets of targets and/or microRNAs from specific species but allow analyzing any input sequence. Among the identified microRNAs, we selected as potential candidates those that were predicted by all tools, except for RNA22 due to its strong inconsistencies with the other predictors.

We obtained a first set of predictions examining all human microRNAs recorded in the release 22 of miRBase and the 3′-UTR sequence of the reference genome of SARS-CoV-2. This first set of 10 microRNAs that potentially binds the SARS-CoV-2 3′-UTR is available in the Appendix A or at OSF (https://osf.io/cujxm/). The list includes a sequence, the 16 nucleotides-long hsa-miR-4279, unlikely to correspond to a real microRNA. To guarantee that chosen sequences correspond to actual human microRNAs, we carried out a second analysis employing those miRBase sequences recognized as high confidence human mature microRNA according to the principles established in Kozomara and Griffiths-Jones [20] (full list available at [21]). No high confidence microRNA was predicted to target the viral 3′-UTR according to miRDB; therefore, we build up this second list using the predictions from IntaRNA, RNA22, RNAhybrid, and STarMir tools. The resulting list of high confidence human mature microRNAs potentially binding SARS-CoV-2 3′-UTR is available in Appendix A and at OSF (https://osf.io/45fne/).

We selected the microRNAs to be tested in vitro according to the values of the principal scores of each predictive tool (miRDB: maximal target score, IntaRNA: energy score (E), RNA22: folding energy and *p*-value, RNAhybrid: minimal free energy of hybridization (mfe), and STarMir: maximal logistic probability (Logit.Prob.)). Any sequence with STarMir Logit.Prob. value below 0.70 was excluded from the selection (consensus between [22] and STarMiR website recommendations). In the analysis of the first set (full miRBase release 22), we also excluded those sequences with a miRDB minimal target score below 60 following the website recommendations. The final selection comprises the top 5 microRNAs from the complete miRBase and the top 5 from the high confidence microRNAs set (Table 1). Details about the predicted bindings—including location, seed sequence complementarity, free energy, target site accessibility, and AU content upstream and downstream—for each selected microRNA are shown in Figure 1.

Throughout these bioinformatics analyses, we have always employed Wuhan SARS-CoV-2 reference genome as target sequence for microRNA prediction. However, multiple variants of SARS-CoV-2 virus are known (see GISAID database), such as mutation D614G [23], which has become prevalent worldwide replacing the original Wuhan variant, or the recent B.1.1.7, 501Y.V2, 484K.V2, 452R.V1, and 452R.R3 lineages, which are now spreading from UK, South Africa, Brazil, California (USA), and India, respectively. To explore whether the 3′-UTR sequence from the Wuhan reference sequence is conserved across variants, we compared it to the 3′-UTR of 63 SARS-CoV-2 sequences covering all GISAID clades and 39 Pangolin Lineages (including the “British”, “South African”, “Brazilian”, and “Indian” variants) sampled between November 2019 and May 2021 (Appendix A). CLUSTALW alignment revealed that the 3′-UTR of all SARS-CoV-2 sequences analyzed here were 100% identical in all but 19 sequences. Changes in these 19 sequences comprised punctual mutations in one (11 sequences) or multiple nucleotides (2 sequences) or gaps up to 10 nucleotides (6 sequences). Gaps at the end of the sequence (by the poly-A tail) were present in the 3 sequences of the “South African variant” (Pango Lineage B.1.351) that were included in the analysis, whereas punctual mutations appear in 2 of the 4 sequences of the “British variant” (Pango Lineage B.1.1.7), in the 2 samples of the “Brazilian variant” (Pango Lineage P.1), in the 2 of the 7 sequences of the “Californian variant” (Pango Lineages B.1.429 and B.1.427), and in the 3 samples of the “Indian variant” (Pango Lineage B.1.617 and sublineages B.1.617.1 and B.1.617.2). An additional change involved a G to U or A mutation in the nucleotide 29,742 that appeared in 4 sequences from 4 different countries sampled from January to September 2020 and may alter the 3′-UTR secondary structure according to Hosseini and McLellan [15]. However, contrary to the potential spreading of the B.1.351, B.1.429, B.1.427, B.1.1.7, P.1, and B.1.617 variants, nucleotide 29,742 mutation does not seem to be spreading in recent times. Therefore, these comparisons indicate that microRNAs identified to target the 3′-UTR of Wuhan reference sequence can also target most SARS-CoV-2 variants although attention should be paid to the effects of the mutations of the British, South African, Brazilian, Californian, and Indian variants.

### 2.2. Human Cell Lines Show Different Endogenous Regulation of SARS-CoV-2 3′-UTR

To in vitro validate the targeting of the predicted microRNAs on SARS-CoV-2 3′-UTR, we carried out reporter assays on a pmiRGLO plasmid construct containing a luciferase reporter under the regulation of SARS-CoV-2 3′-UTR (pmiRGLO-3′-UTR-COVID-19). Before carrying out these assays, we transfected the pmiRGLO-3′-UTR-COVID-19 construction alone in SH-SY5Y, HeLa, HEK293T, and Caco-2 human cell lines to explore the endogenous regulation of the plasmid expression. As shown in Figure 2, this analysis indicated the existence of particularly active endogenous mechanisms regulating the expression of the SARS-CoV-2 3′-UTR in HeLa cervix epithelial cells (46% expression compared to control construct pmiRGLO without subcloned plasmid, *p* < 0.01) and to a lesser degree in Caco-2 colon epithelial cells (77%, *p* < 0.05) and SH-SY5Y neuroblastoma-like cells (75%, *p* = 0.247). On the contrary, HEK293T kidney cells did not show appreciable reductions in the construct expression (102%, *p* = 0.092). In summary, results suggest the existence of endogenous post-regulation on viral 3′-UTR controlling the expression of gene reporter with different activity depending on the cell type. These results led us to select HEK293T as the most appropriate cell line for the reporter assays due to its lack of endogenous regulation.

### 2.3. MicroRNAs Can Target the 3′-UTR Sequence of SARS-CoV-2

To determine whether any of the 10 microRNAs selected from the bioinformatics predictions do effectively target SARS-CoV-2 3′-UTR, we performed a dual-luciferase reporter assay co-transfecting human HEK293T cells with pmiRGLO-3′-UTR-COVID-19 plasmid and mimics of the microRNA candidates or cel-miR-67 negative control. In the analyses, we also included the central nervous system-specific hsa-miR-138-5p—a human microRNA currently under study in our laboratory for an unrelated research—as a negative control that did not meet the selection criteria (without binding sites in the SARS-CoV-2 3′-UTR according to miRDB and IntaRNA, although with a STarMir Logit.Prob. of 0.713).

As shown in Figure 3A, luciferase activity measurements indicate that the negative control microRNA did not affect pmiRGLO-3′-UTR-COVID-19 expression (94% vs. pmiRGLO-3′-UTR-COVID-19, n.s.), so we used it as a reference to evaluate the effects of the microRNA mimics. Concerning the microRNAs under analysis, the results revealed that transfection of their respective mimics had variable effects on the expression of SARS-CoV-2 3′-UTR, from reductions above the 50% (e.g., hsa-miR-4433-3p) to no appreciable differences such as after transfection of hsa-miR-466-5p, hsa-miR-5088-5p, or hsa-miR-5010-5p mimics (Figure 3A). Transfection with mimics of hsa-miR-3941, hsa-miR-128-1-5p, hsa-miR-4433b-3p, and hsa-miR-365b-5p caused statistically significant reductions above the 33% of the values measured after transfection with the negative control (paired Student’s *t* test, one-tailed *p*-value < 0.05 after Bonferroni correction). Surprisingly, hsa-miR-138-5p led to a highly significant reduction of 41.8% (*p* < 0.05), the highest observed among all the tested microRNAs (Figure 3A). In order to exclude that reduction in luciferase activity was due to cytotoxicity of transfection, we performed parallel MTT assays (Figure 3B), which indicated that 24 h of mimic transfection did not markedly reduce cell viability (i.e., above 10% relative to DharmaFECT effect) except when transfecting hsa-miR-4433-3p (18.67% reduction) and hsa-miR-365b-5p (22.24% reduction), both microRNAs with significant effects on SARS-CoV-2 3′UTR. None of the observed differences was statistically significant according to paired *t*-tests.

To discard that the effects of the microRNAs on the expression of pmiRGLO-3′-UTR-COVID-19 result not from the effect on the viral 3′-UTR sequence cloned in the plasmid but on other regions of the plasmid sequence, we evaluated the effects of the 5 selected microRNAs that significantly reduced pmiRGLO-3′-UTR-COVID-19 expression on the empty pmiRGLO construct. Luciferase assay did not reveal significant effects of hsa-miR-3941, hsa-miR-128-1-5p, hsa-miR-365b-5p, and hsa-miR-138-5p on the empty pmiRGLO construct (Figure 4), thus supporting that the effects of these microRNAs on the pmiRGLO-3′-UTR-COVID-19 expression were not due to luciferase gene post-transcriptional regulation. Conversely, hsa-miR-4433b-3p reduced firefly/Renilla luminescence ratio of the empty pmiRGLO plasmid (Figure 4), and therefore, we cannot rule out that the observed reduction in pmiRGLO-3′-UTR-COVID-19 expression (Figure 3A) results from unspecific effects of this microRNA on the luciferases mRNAs.

To further characterize the potential of those microRNAs showing the best properties for viral 3′-UTR regulation (hsa-miR-3941, hsa-miR-128-1-5p, hsa-miR-365b-5p, and hsa-miR-138-5p), we employed COVID CG open resource to search for sequence variations in the viral 3′-UTR, which may affect microRNA binding. Confirming our initial analyses of the variations in the 3′-UTR sequence, the reference sequence is preserved in more than 80% of the sampled specimens. As illustrated in Table 2, the binding sites for hsa-miR-3941 (nt 29,678-29,696), hsa-miR-138-5p (nt 29,706–29,744), hsa-miR-365b-5p (nt 29,783–29,805), and hsa-miR-128-1-5p (nt 29,821–29,839) in the SARS-CoV-2 3′-UTR presented mutations that involve nucleotides binding to the microRNAs (marked in red), even at the seed region (in red bold). Mutated nucleotides were particularly frequent in the binding region of hsa-miR-138-5p (mutations observed in 18 nucleotides), whereas hsa-miR-3941 and hsa-miR-128-1-5p only present mutations in 4 and 2 nucleotides, respectively. Most mutations were present in less than a 1% of the sampled sequences with exceptions corresponding to nucleotides 29,692 (1.4%), 29,732 (1.1%), 29,734 (5.1%), and 29,796 (7.6%) in the binding regions of hsa-miR-3941, hsa-miR-138-5p, and hsa-miR-365b-5p, respectively. However, although these mutations were still being recorded in January and February 2021 samples, their frequencies were in clear decline and did not reach a 2% of February samples (no data are provided for mutations in nt 29,796). So, neither mutations potentially affecting the binding of the selected microRNAs have spread during the pandemic nor have accumulated in the last months.

## 3. Discussion

SARS-CoV-2 is a novel and exceptionally infective coronavirus that has spread to more than 220 countries across the world during 2020 and 2021. Various authors have proposed that endogenous cellular microRNAs may help fight COVID-19 infection and have predicted putative binding sites for human microRNAs in the SARS-CoV-2 genome using in silico tools [11,12,13,14,15,16,17]. In the present study, we have gone one step ahead, validating bioinformatics predictions through gene reporter assays in human cell cultures. We have focused our analysis on the viral 3′-UTR, a frequent target region for microRNAs that can block viral replication (e.g., [24]).

Our screening of microRNA targeting SARS-CoV-2 3′-UTR according to the scores from 5 highly cited predictive tools (miRDB, IntaRNA, RNA22, RNAhybrid, and STarMir) provided us with a selection of 10 putative microRNAs. The list includes hsa-miR-4775, already predicted to target SARS-CoV-2 3′-UTR by Yousefi and colleagues and which also regulates Smad7 protein in the TGF-beta pathway, thus modulating the immune response after infection [25]. The list also includes hsa-miR-1307-3p, predicted to regulate SARS-CoV-2 expression due to its high binding affinity to viral genome [17,26,27,28,29], its regulation of H5N1 expression [30,31], and its conservation across species, including humans [32].

We validated the effects of the 10 microRNAs selected from the computational screening using a luciferase gene reporter assay with a plasmid that expresses firefly luciferase gene containing the full SARS-CoV-2 3′-UTR. The results from gene reporter assays initially confirmed the targeting by 4 of the 10 predicted microRNAs. However, we discarded one of them (hsa-miR-4433b-3p) because it has an unspecific effect on the expression of the pmiRGLO-3′-UTR-COVID-19 plasmid by inhibiting luciferase expression of an empty pmiRGLO plasmid (without the subcloned 3′-UTR). The remaining list of effective microRNAs comprises hsa-miR-3941, with two canonical binding sites (one 8mer and one 7mer-A1) in the viral 3′-UTR and the highest scores of miRDB, together with hsa-miR-128-1-5p and hsa-miR-365b-5p, both without canonical binding regions in the seed region and some of the lower scores in the predictive tools (Table 1). Surprisingly, the list of microRNAs effectively targeting SARS-CoV-2 3′-UTR does not include hsa-miR-4775 and hsa-miR-1307-3p, both with binding sites according to various computational target prediction analyses [17,25,26,27,28,29,32]. Conversely, the list comprises hsa-miR-138-5p, a microRNA that we included in the analysis as a negative control. Indeed, no binding sites for hsa-miR-138-5p in the 3′-UTR were predicted by miRDB, IntaRNA, and RNA22 although the binding site illustrated in Figure 5 was predicted by STarMir with a Logit.Prob. above the limit established in this study (0.7).

Gene reporter assays indicate that hsa-miR-3941, hsa-miR-128-1-5p, hsa-miR-365b-5p, and hsa-miR-138-5p target the viral 3′-UTR and, therefore, they may control viral pathogenicity. To our knowledge, besides bioinformatics predictions, nothing has been published relating any of these microRNAs and COVID-19. Indeed, very little has been published on the relationships between any microRNA and SARS-CoV-2 beyond the results from bioinformatics analyses. However, published studies on microRNA function and activity as well as on their validated targets [33,34] can provide evidence supporting or dismissing their therapeutic potential. In this respect, there is evidence on the activity of hsa-miR-3941 in the respiratory system associated with a potential antiviral mechanism. Hsa-miR-3941 appears downregulated in children with acute pneumonia and in vitro analyses have revealed that it inhibits the phosphoinositol 3-kinase (PI3K)/serine-threonine kinase (AKT) signaling response to protect A549 human alveolar basal epithelial cells from injury [35]. According to Kindrachuk and colleagues [36], the PI3K/AKT signaling response promotes the replication of MERS-CoV and other viruses, whereas their inhibition is capable of blocking viral replication in vitro, probably through a mechanism related to the Warburg effect [37]. According to miRTarBase, hsa-miR-3941 includes *AKAP11* and *BEX4* among its validated targets, the former reducing its phosphorilation state and the latter its expression after SARS-CoV-2 infection in vitro [38,39]. However, their relation to SARS-CoV-2 infection remains unclear. Therefore, the therapeutic upregulation of hsa-miR-3941 may combine the direct blockage of viral expression through 3′-UTR targeting with the inhibition of PI3K/AKT pathway to reduce viral infection.

Among the extensive bibliography on miR-138-5p, much concerns cancer processes and central nervous system functioning. However, a few references also relate it to COVID-19, in particular, bioinformatics predictions indicate that miR-138-5p may target SARS-CoV-2 genes [40,41] as well as IL6 and IL-8 cytokines, two major actors in the COVID-19-induced “cytokine storm” associated with poor outcomes [42]. In addition, reports by Pan and colleagues [43,44] indicate that miR-138-5p targets herpes simplex virus mRNA and is key to promote viral latency and reduce morbidity and mortality in mice. Moreover, hsa-miR-138-5p has validated targets—*DEK*, *EED*, and *EZH2*—among genes included in GO term viral process (GO:0016032 from Gene Ontology) [45,46]. The histone H3K27 methyltransferases EZH2 is particularly interesting since its inhibitors seem capable to suppress infection with DNA and RNA viruses according to Arbuckle and colleagues [47]. As a whole, evidence suggests that hsa-miR-138-5p may be a good candidate for therapeutics, capable to regulate both the viral replication and the associated inflammatory response, as it does in herpes simplex virus.

Much less is known about miR-365b-5p and miR-128-1-5p. According to Pan and colleagues [48], overexpression of miR-128-1-5p induces tight junction structural damage and increases cellular and paracellular permeability, resembling the early-stage processes of COVID-19 infection [49] and dismissing the potential antiviral effects of this microRNA. Evidence compiled in miRTarBase indicates that miR-128-1-5p has 3 validated targets—*AC1N1, AP2S1, COLGALT1*—among the proteins changing after viral infection according to H2V database. One such change involves the interaction of SARS-CoV-2 nsp1 protein with Procollagen galactosyltransferase 1 (*COLGALT1* gene) to potentially antagonize the innate immune response [50]. Another COVID-19-induced change involves the deubiquitination of AP2S1 [51], a subunit of AP2 adapter protein complex for clathrin also included within the Gene Ontology term Viral Process, which may contribute to the viral entrance. GO term Viral Process also includes *EPHA2*, another miR-128-1-5p validated target employed by different viruses to enter the cells [52], although direct evidence in SARS-CoV-2 is still lacking. Therefore, the available evidence about miR-128-1-5p is somehow contradictory and is difficult to evaluate its therapeutic potential without further analyses. Regarding miR-365b-5p, available information is even scarcer. According to the miRTarBase database of validated targets, it targets Caveolin-2, a component of lipid rafts where RNA virus such as HCV replicate [53], which may support some antiviral activity.

Although coronaviruses are known to have genetic proofreading mechanisms and SARS-CoV-2 seems to have lower sequence diversity than virus such as HIV [23], multiple variants have emerged during the course of the pandemic. Some of these variants—e.g., lineages B.1.1.7, 501Y.V2, 484K.V2, 452R.V1, and 452R.R3 from UK, South Africa, Brazil, California (USA), and India, respectively—show increased human-to-human transmissibility and have spread in different regions of the world, potentially challenging the efficacy of the available vaccines [54]. Although many variants (including British, South African, Brazilian, Californian, and Indian variants) have mutations in the 3′-UTR that could affect microRNAs binding, our analyses indicate that only the Indian variant, among the major variants, presents mutations in the binding sites between the microRNAs here identified and SARS-CoV-2 mRNA (nt 29,837 for miR-128-1-5p and nt 29,734 for miR-138-5p). Moreover, none of the mutations in the binding sites of miR-3941, miR-128-1-5p, miR-365b-5p, and miR-138-5p is present in more than a 1% of all samples sequenced by February 2021.

It has been described that a mutation in nucleotide 29,742 modifies the secondary structure of the 3′-UTR [15] and may challenge microRNA accessibility to the 3′-UTR binding sites. However, this mutation remains scarce—it is present in a 0.3% of all GISAID deposited sequences and a 0.25% of those deposited in January and February 2021 according to COVID CG resource—and does not affect the binding affinity of hsa-miR-3941, hsa-miR128-1-5p, hsa-miR-365b-5p, and hsa-miR-138-5p according to IntaRNA and RNAhybrid predictive tools (both tools explicitly consider secondary structure and target accessibility for their scores). Moreover, some studies have hypothesized that microRNAs regulating SARS-CoV-2 may also target SARS and MERS coronavirus [12,55]. Full sequence similarity between SARS-CoV-2 with SARS and MERS is 79% and 50%, respectively [10]. Although their 3′-UTR show strong size differences (i.e., SARS-CoV-2 is 196 nt long, whereas both SARS and MERS are around 1000 nucleotides), a 94% and a 33% of the SARS-CoV-2 3′-UTR sequence is preserved in the SARS and MERS, respectively. Accordingly, 3 of the microRNAs targeting SARS-CoV-2 (all but hsa-miR-138-5p) preserve the same binding sites in SARS 3′-UTR, whereas they are completely absent from the 3′-UTR of MERS according to IntaRNA (data not shown). Recently, Mohammadi-Dehcheshmeh and colleagues [16] have identified microRNAs that may target leading and transcription regulatory sequences at the 5′-UTR of SARS-CoV-2 and other coronaviruses. According to the authors, these microRNAs may reduce viral RNA replication and be key to lower viral load in the infected cells. The identified microRNAs include only one human microRNA, namely, hsa-miR-5004-3p, which, interestingly, presents binding sites in the 3′-UTR according to RNAhybrid (mfe = −25.9) and STarMiR (Logit.Prob = 0.749). Although we have not analyzed this microRNA because it did not fulfill all the criteria established for the in silico analyses, hsa-miR-5004-3p would merit further attention due to its potential targeting of both 5′- and 3′-UTRs.

In addition to the main focus of this study, the development of a SARS-CoV-2 3′-UTR construction and its analysis in different cell lines provided interesting information on the different susceptibility to infection among cell types described, e.g., in the study of Chen and colleagues [56]. Although the main symptoms of SARS-CoV-2 infection have been found in respiratory tract, viral infection has been also found in other organs, including kidney [57], intestine [58,59], and brain [60,61] probably related to their expression of ACE2 receptor and TMRPS2 [62]. Studies in cell lines revealed strong differences in infection susceptibility and confirmed that the expression of ACE2 and TMRPS22 is a major determinant of infectivity [56]. However, the infection susceptibility remained low in cell lines such as HeLa even after forced overexpression of both molecules, claiming for the existence of additional regulatory mechanisms. Our analyses on the endogenous regulation of SARS-CoV-2 3′-UTR in SH-SY5Y, HeLa, HEK293T, and Caco-2 human cell lines, and particularly its strong downregulation in the HeLa cells, suggest that this may be one of such mechanisms and point to microRNAs as potential regulators. Unfortunately, the expression of none of the microRNAs identified here to target the viral 3′-UTR correlates with the observed endogenous regulation of the 3′-UTR (data not shown). Further studies are required to verify the existence of endogenous regulation of the SARS-CoV-2 3′-UTR and identify the underlying mechanism.

## 4. Materials and Methods

### 4.1. SARS-CoV-2 Sequences

All bioinformatics and cloning analyses performed in this study were based on the 3′-UTR of SARS-CoV-2 reference sequence (NC_045512.2, from Wuhan, China). We obtained reference sequence and any other sequence information employed through this study from NIH’s GenBank [63] and GISAID [64,65]. The full list of viral sequences employed in this article is available in the attached Appendix A as well as in OSF (https://osf.io/gtx2j/). We obtained sequence information on human microRNAs from miRBase database [20,66].

### 4.2. Computational Prediction

We carried out an in silico screening to predict microRNA response elements (MREs) in the 3′-UTR of SARS-CoV-2 sequence, according to the principles of microRNA target prediction described in [19] and listed in Table 3.

To this aim, we analyzed the full list of human microRNAs from miRBase v22 [66] and a subset of high confidence human mature microRNAs [21] (details at [69]) together with the reference sequence of the 3′-UTR of SARS-CoV-2 (NC_045512.2, from nt 29,675 to 29,903) as target sequence using the following 5 predictive tools (Table 3), freely available online:miRDB [70,71]: we employed version 6.0 (updated in July 2019) with miRTarget version 4 and miRBase version 22 human microRNA lists. We employed sequence of SARS-CoV-2 to search for human microRNAs targeting on it, using default settings.IntaRNA ([72], version 2.0) [53]: we selected 10 interactions per RNA pair (microRNA/mRNA SARS-CoV-2 3′-UTR) with a minimal number of 6 base pairs in the seed region. No other parameters were modified. We exported the interactions from lower to higher free energy values and the 100 interactions with the lower E in CSV format.RNA22 ([73], version 2) [74]: we selected a seed size of 7 nucleotides and a maximum of 1 unpaired base. No other parameters were modified. The full list of microRNA-mRNA pairs with folding energy and *p*-value data was selected.RNAhybrid [75,76]: we selected 10 binding sites per RNA pair. No other parameters were modified. The list of the microRNA-mRNA pairs and their mfe and *p*-value data was selected.STarMir [77,78]: we manually introduced the microRNA lists in groups of 20, the NCBI genome ID of SARS-CoV-2 mRNA, and its 3′-UTR sequence. As STarMir require information on the CDS start and end points in the sequence, we included one additional nucleotide from the 5′end of the 3′-UTR, which served as both the start and end nucleotide of the CDS. The predictions, including Logit.Prob. values, were obtained.

### 4.3. Sequence Alignments and Analysis of Variants

We employed CLUSTALW [79,80] to align the 3′-UTR of the mRNA of SARS-CoV-2 sequence from Wuhan with viral sequences from different variants and origins. Besides, we further explored variability in the 3′-UTR sequence of SARS-CoV-2 strains around the world employing COVID CG open resource [81,82].

### 4.4. Subcloning of SARS-CoV-2 3′-UTR Sequence

We obtained the wild-type viral SARS-CoV-2 3′-UTR sequence without poly-A tail (196 nt, from nt 29,675 to 29,871 in the reference sequence) from the assembly of two opposite overlapping primers, followed by amplification by PCR. We subcloned the PCR product into a T vector plasmid (pGEM-T-easy, Promega, Madison, WI, USA), and we validated the sequence by DNA sequencing (T7p and SP6). Then, we inserted the SARS-CoV-2 3′-UTR sequence into the pmirGLO Dual-luciferase miRNA Target Expression Vector (Promega, a scheme on the reporter construct is available at [83]) between the SacI and SalI restriction sites using the FastDigest restriction enzymes (Thermo Scientific, Waltham, MA, USA). Finally, we confirmed the correct sequence and orientation of SARS-CoV-2 3′-UTR in the resulting plasmid (pmiRGLO-3′-UTR-COVID-19) by DNA sequencing using a specific forward 3′ end luciferase primer. Primers used for the preparation of the SARS-CoV-2 3′-UTR and DNA sequencing are listed in Table 4.

### 4.5. Cell Culture

We cultured HEK293T human embryonic kidney cells (cat#: CRL-1573, ATCC) and Caco-2 human colon epithelial cells (cat#: HTB-37, ATCC) in Dulbecco’s modified Eagle’s medium (Gibco) supplemented with 10% fetal bovine serum (FBS; Gibco), 1% penicillin/streptomycin (Gibco), and 1% glutamine (Gibco); HeLa human cervix epithelial cells (cat#: CCL-2, ATCC) in Minimum Essential Medium (Gibco) supplemented with 10% FBS, 1% penicillin/streptomycin, 1 mM sodium pyruvate (Gibco), 1% glutamine, and non-essential amino acids (Gibco); and SH-SY5Y human neuroblastoma-like cells derived from a bone marrow metastatic tumor (cat#: CRL-2266, ATCC) in a 1:1 combination of Minimum Essential Medium and Ham’s F-12 nutrient mixture (Gibco) supplemented with 10% FBS, 1% penicillin/streptomycin, 1 mM sodium pyruvate, and non-essential amino acids. The cells were cultured at 37 °C in a humidified incubator containing 5% CO_2_.

### 4.6. Dual Luciferase Reporter Assays

We first tested the endogenous regulation of pmiRGLO-3′-UTR-COVID-19 plasmid in the selected cell lines. We grew cells to 70% confluence in white 96-well plates and transfected them with 300 ng/well of pmiRGLO-3′-UTR-COVID-19 or empty pmiRGLO (without subcloned 3′-UTR) employing TurboFect Transfection Reagent (cat#: R0531, Thermo Fisher Scientific). Then, 24 h later, we evaluated plasmid gene expression by measuring firefly and Renilla luciferase activities using the Dual-GLO luciferase assay system (Promega) in an Infinite M200 plate reader (Tecan) according to the manufacturer’s protocol. Background values of firefly and Renilla luciferase luminescence from the cells treated only with TurboFect Transfection Reagent were subtracted to the values of the cells transfected with pmiRGLO-3′-UTR-COVID-19 or empty pmiRGLO. Results were expressed as firefly/Renilla ratio to normalize firefly reporter activity to Renilla load control values. All experiments were performed in triplicate and repeated at least three times.

To validate the targeting of the predicted microRNAs on SARS-CoV-2 3′-UTR, we cultured HEK293T cells to 70% confluence in white 96-well plates. Then, we co-transfected cells using DharmaFECT Duo Transfection Reagent (DharmaconTM) with either 50 nM of the specific microRNA mimic (Cherry-pick Library, DharmaconTM) or 50 nM negative control mimic (synthesized by Eurogentec from the cel-miR-67 mimic sequence, without known targets in the human genome; MIMAT0000039) and 200 ng/well of pmiRGLO-3′-UTR-COVID-19 or empty pmiRGLO. Then, 24 h later, we measured firefly and Renilla luciferase activities as described above. All experiments were performed in triplicate and repeated at least three times.

### 4.7. MTT Assay

To evaluate the toxicity of microRNA mimics, we cultured HEK293T cells overnight at a density of 10,000 cells per well in transparent 96-well plates and co-transfected with either 50 nM microRNA mimics or 50 nM negative control mimic and 200 ng/well of pmiRGLO-3′-UTR-COVID-19 using DharmaFECT Duo Transfection Reagent. After 24 h, we incubated cultures with (3-(4,5-dimethylthiazol-2-yl)-2,5-diphenyltetrazolium bromide) tetrazolium (MTT) in the medium at a final concentration of 0.5 mg/mL for 30 min. We dissolved formazan crystals by addition of 100 μL/well of HCl:isopropanol (1:500) solution followed by measurement of the absorbance at 570 nm in a plate reader (Infinite M200, Tecan). We used absorbance at 690 nm for background subtraction.

### 4.8. Data Analysis

Graph plots show the mean and standard deviation (SD) of, at least, 3 independent experiments and with experimental triplicates each one. We tested the statistical significance of the treatment effects using paired Student’s *t* test. Bonferroni correction for multiple testing was applied when required. We identified outliers using Grubbs’ test. We conducted all statistical analyses using Prism Software 5 (GraphPad Software Inc., La Jolla, CA, USA) except as otherwise specified. We considered differences statistically significant when *p* ≤ 0.05.

## 5. Conclusions

The combined results from our bioinformatics and gene reporter analyses together with the available bibliography indicate that microRNAs, particularly, hsa-miR-3941 and hsa-miR-138-5p, could be helpful to reduce viral pathogenicity of major current variants of SARS-CoV-2 or even SARS, in the case of hsa-miR-3941. This study is highly preliminary and further analyses—from reporter assays of additional candidates (e.g., hsa-miR-5004-3p [16]) to microRNA expression and effects in COVID-19-infected cells and animals—are needed to validate the therapeutic potential of this approach. To our surprise, very little has been advanced in this respect judging from the published records despite the urgent need for COVID-19 treatments and the recent technological advances for RNA-therapeutics (e.g., RNA nanoformulations (see review [84]), already employed during the pandemic (i.e., Pfizer and Moderna vaccines) and which should allow a successful and safe delivery of microRNAs mimetics or even antagonists (as has been proposed for miR-122) to infected cells. We hope that the present study will contribute to foster the exploration of this approach.

## Figures and Tables

**Figure 1 ijms-22-06094-f001:**
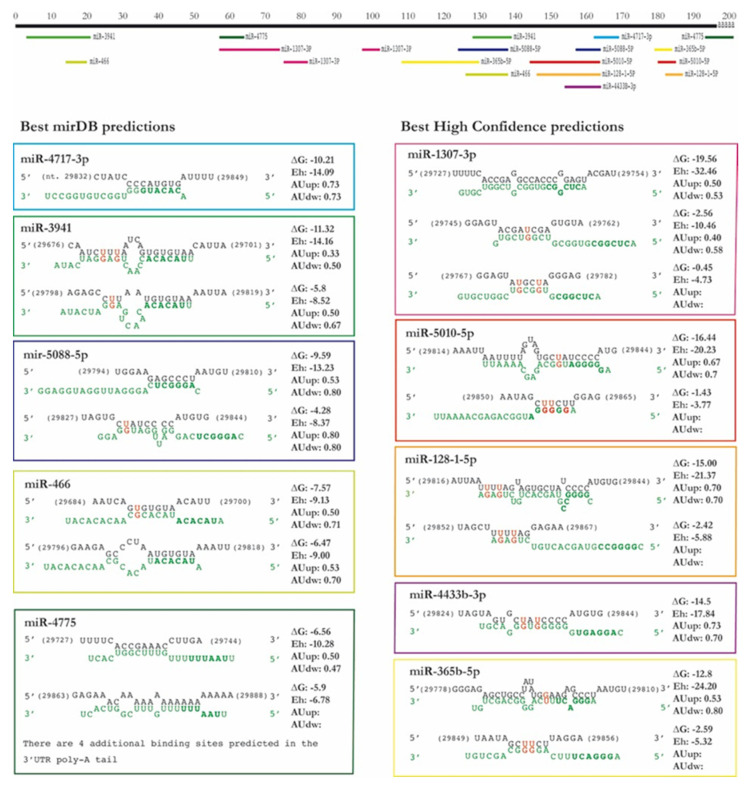
Predicted bindings of the selected microRNAs in the SARS-CoV-2 3′-UTR. The figure summarizes the information provided by the different predictive tools on location (derived from IntaRNA), sequence complementarity (IntaRNA and STarMir), free energy (IntaRNA), target site accessibility (IntaRNA and RNAhybrid), and AU content upstream and downstream (provided by STarMir). The viral 3′-UTR sequence is written in black, whereas the human microRNA sequence appears in green type, microRNA seed regions appear in bold type, and GU unions are shown in red. The position of the binding regions in the 3′-UTR (without poly-A, in black) for each microRNA appears represented in the upper part of the figure.

**Figure 2 ijms-22-06094-f002:**
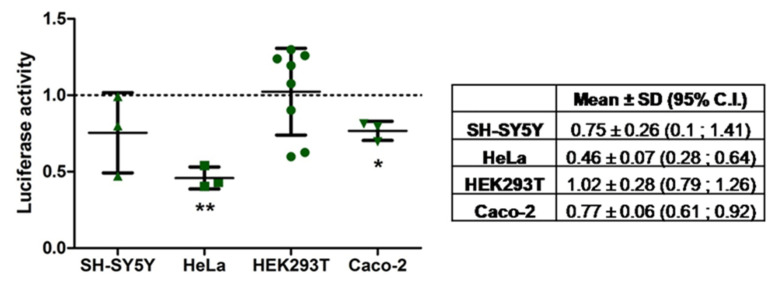
Endogenous regulation of pmiRGLO-3′-UTR-COVID-19 plasmid expression in different human cell lines. Dot plot represents the firefly/Renilla luminescence ratio of the 3′-UTR construct in each cell line and experiment normalized to the ratio from pmiRGLO construct for the same cell line and experiment. Overlaid line plot represents the mean ± SD of, at least, three experiments, and the dotted line represents the reference value for pmiRGLO construct (equal to 1 after normalization). An extended figure with all experimental replicate data points is available in Appendix A. The associated table summarizes the mean ± SD with 95% of C.I. of three independent experiments except for HEK293T, which comprises eight independent experiments, including the data from the reporter assays described below. In HeLa and Caco-2 cells, the 95% C.I. did not include the reference value of pmiRGLO indicating a significant (*p* < 0.05) regulation of the SARS-CoV-2 3′-UTR. SH-SY5Y and Caco-2 cells showed a moderate endogenous regulation of the expression of the SARS-CoV-2 3′-UTR. HeLa cells reduced luciferase activity of the SARS-CoV-2 3′-UTR to more than 50% compared to the pmiRGLO. In HEK293T cells, the luciferase activity between both plasmids was very similar, so there was no endogenous regulation by these cells. * indicates a *p*-value < 0.05 and ** *p*-value < 0.01.

**Figure 3 ijms-22-06094-f003:**
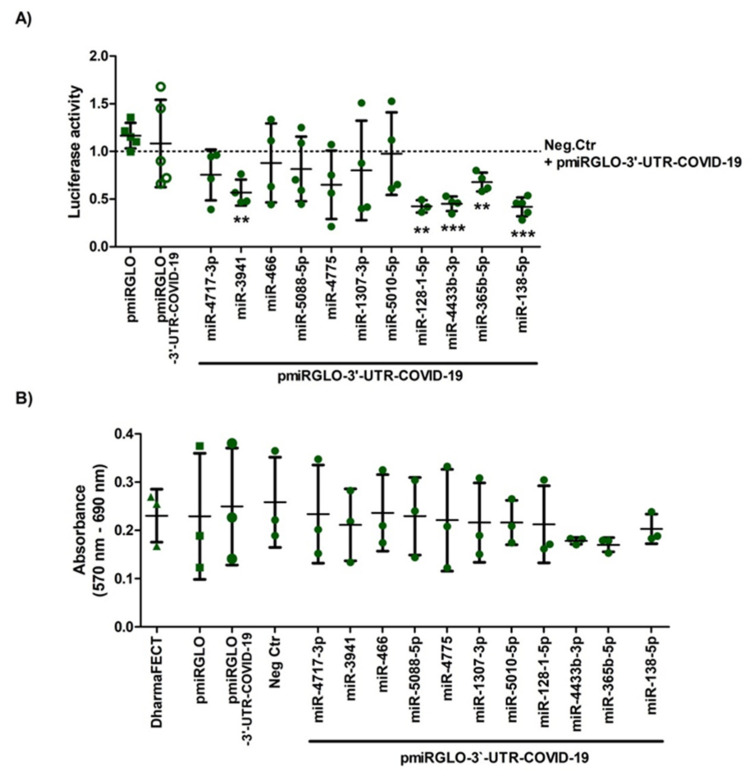
Regulation of pmiRGLO-3′-UTR-COVID-19 plasmid expression by selected microRNAs in HEK293T cells and cell viability after transfection. Cells were transiently co-transfected with control or reporter constructs and selected microRNA mimics or the negative control microRNA. (**A**) Firefly/Renilla luminescence ratio showed that hsa-miR-3941, hsa-miR-128-1-5p, hsa-miR-4433b-3p, and hsa-miR-365b-5p, as well as hsa-miR-138-5p, significant decreased the expression of pmiRGLO-3′-UTR-COVID-19 related to the expression of negative control mimic (paired Student’s *t* test, one-tailed *p*-value < 0.05 after Bonferroni correction). ** indicates a *p*-value < 0.01 and *** *p*-value < 0.001; (**B**) On the other hand, MTT assay data showed that empty pmiRGLO or pmiRGLO-3′-UTR-COVID-19 plasmids and mimics did not produce any significant cytotoxic effect related to DharmaFECT after 24 h of co-transfection (paired Student’s *t* test). Graph represents the mean ± SD of, at least, three experiments. An extended figure with all experimental replicate data points is available in Appendix A.

**Figure 4 ijms-22-06094-f004:**
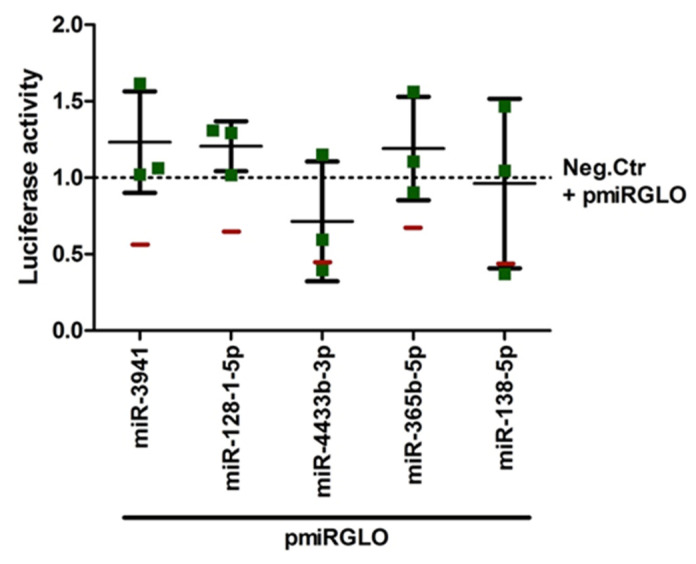
Regulation of empty pmiRGLO plasmid expression by selected microRNAs in HEK293T cells. The cells were transiently co-transfected with control construct and selected microRNA mimics or the negative control microRNA. Firefly/Renilla luminescence ratio showed that hsa-miR-3941, hsa-miR-128-1-5p, hsa-miR-365b-5p, and hsa-miR-138-5p did not regulate the luciferase expression of empty pmiRGLO construct (control plasmid), which show values similar to the negative control microRNA effect, so these microRNAs did not alter luciferase gene post-transcriptional regulation. Red lines representing the mean luciferase values of the pmiRGLO-3′-UTR-COVID-19 plasmid after treatment with each microRNA mimic are included as reference for comparison. Graph represents the mean ± SD of, at least, three experiments performed in triplicates. An extended figure with all experimental replicate data points is available in Appendix A.

**Figure 5 ijms-22-06094-f005:**
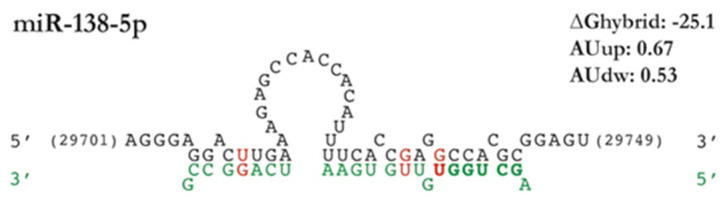
Predicted binding site of hsa-miR-138-5p in the SARS-CoV-2 3′-UTR. The figure summarizes the information provided by STarMir and RNAhybrid on location, sequence complementarity, free energy, target site accessibility, and AU content upstream and downstream. The viral 3′-UTR sequence is written in black, whereas the human microRNA sequence appears in green type, microRNA seed regions appear in bold type, and GU unions are shown in red.

**Table 1 ijms-22-06094-t001:** Prediction scores of selected microRNAs. The table shows the values of miRDB target score, IntaRNA energy score (E), RNA22 folding energy and *p*-value, RNAhybrid minimum free energy (mfe), and STarMir logistic probability (Logit.Prob.) of the 10 chosen microRNAs.

	microRNA	miRDB	IntaRNA	RNA22	RNAhybrid	STarMir
		Target Score	Energy Score	Folding Energy (*p*-Value)	Minimum Free Energy	Logit. Prob.
**Complete miRBase 22**	hsa-miR-4717-3p	68	−10.21	−16 (0.33)	−26.3	0.898
	hsa-miR-3941	84	−11.32		−19.2	0.805
	hsa-miR-466	81	−7.57		−14.7	0.756
	hsa- miR-5088-5p	60	−9.59		−21.8	0.803
	hsa-miR-4775	78	−6.56		−15.7	0.719
**High confidence miRBase 22**	hsa-miR-1307-3p		−19.56	−31.1 (0.224)	−37.6	0.761
	hsa-miR-5010-5p		−16.44		−28.7	0.769
	hsa-miR-128-1-5p		−15		−30	0.879
	hsa-miR-4433b-3p		−14.5		−26.5	0.86
	hsa-miR-365b-5p		−12.8		−34.2	0.831

**Table 2 ijms-22-06094-t002:** Sequence variability in the binding sites of hsa-miR-3941, hsa-miR-138-5p, hsa-miR-365b-5p, and hsa-miR-128-1-5p. The table indicates the mutations and frequencies in each nucleotide of SARS-CoV-2 3′-UTR. The analysis is restricted to the 3′-UTR sequence of the best binding sites for each microRNA according to the prediction algorithms. Mutations in nucleotides complementary to the microRNAs are represented in red, whereas those complementary to the seed region are marked in bold red. Data are extracted from COVID CG, accessed on 22 February 2021 (511,596 sequences analyzed).

					Nowadays	
	Nucleotide	Frequency (%)	Mutation	Dates	Yes/No	Where
**hsa-miR-3941**	29,679	0.5	T	July 20–Jan 21	No	
	29,686	0.3	T	Sept 20–Jan 21	No	
	**29,690**	0.1	T	Jan 21	No	
	**29,692**	1.4	T	May 20–Jan 21	No	
**hsa-miR-138-5p**	29,706	0.3	T	Jun 20–Feb 21	Yes	All
	29,708	0.1	T	Mar 20–Feb 21	Yes	All
	29,710	0.5	C	Apr 20–Feb 21	Yes	The United States
	29,711	0.1	T	Mar 20–Feb 21	Yes	The United States and Europe
	29,717	0.1	A	Apr 20–Feb 21	Yes	Europe
	29,721	0.1	T	Jun 20–Jan 21	No	
	29,726	0.2	-	Oct 20–Feb 21	Yes	Europe
	29,730	0.4	T/G	May 20–Feb 21	Yes	Europe and The United States
	29,732	1.1	A/G	Jul 20–Feb 21	Yes	Europe
	29,733	0.2	T	Apr 20–Feb 21	No	Canada
	29,734	5.1	T/G/A/C	Apr 20–Feb 21	Yes	The United States and Europe
	29,736	0.1	T	May 20–Feb 21	Yes	All
	29,737	0.3	C	Jun 20–Feb 21	Yes	The United States and Europe
	**29,738**	0.1	T	May 20–Feb 21	Yes	The United States and Europe
	**29,740**	0.3	A	Sept 20–Feb 21	Yes	Europe
	**29,741**	0.4	T	Sept 20–Feb 21	Yes	The United States and Europe
	29,742	0.7	T/A	Mar 20–Feb 21	Yes	The United States and Europe
	**29,743**	0.4	T	Apr 20–Feb 21	Yes	All
**hsa-miR-365b-5p**	29,784	0.4	T	May 20–Feb 21	Yes	The United States
	29,785	0.2	A	Jun 20–Feb 21	Yes	Europe
	29,791	0.8	C/G/T			
	29,796	7.6	C/G/A			
	29,797	0.3	T	Dec 20–Feb 21	Yes	Europe
	**29,798**	0.3	C	Jan 21–Feb 21	Yes	Europe
	**29,799**	0.4	-/C	Dec 20–Feb 21	Yes	Europe
	**29,803**	0.1	T	Oct 20	No	
**hsa-miR-128-1-5p**	29,825	0.1	T	Jul 20–Jan 21	No	

**Table 3 ijms-22-06094-t003:** Principles used for computational microRNA prediction on which miRDB, IntaRNA, RNA22, RNAhybrid, and STarMir are based. Based on [19,67,68].

Principle	miRDB	IntaRNA	RNA22	RNAhybrid	STarMiR
Seed sequence complementary	X	X	X	X	X
Free energy	X	X	X	X	X
G-U wobble	X	X	X	X	X
Evolutionary conservation status	X		X	X	
3′-UTR compensatory binding	X				X
Target-site accessibility	X	X		X	X
Target-site abundance		X			
Local AU flanking content	X				X
Machine learning	X				X
Pattern-based approach			X	X	

**Table 4 ijms-22-06094-t004:** Primers used for subcloning of SARS-CoV-2 3′-UTR.

Primers	Sequences
Forward SARS-CoV-2 3′-UTR SacI	5′ctcgagctctaacaatctttaatcagtgtgtaacattagggaggacttgaaagagccaccacattttcaccgaggccacgcggagtacgatcgagtgtacagtgaacaatgctagggaga3′
Reverse SARS-CoV-2 3′-UTR SalI	5′ctcgtcgactgtcattctcctaagaagctattaaaatcacatggggatagcactactaaaattaattttacacattagggctcttccatataggcagctctccctagcattgttcactgt3′
Forward pmiRGLO sequencing	5′caagaagggcggcaagatcg3′

## Data Availability

All information obtained while conducting the present analysis is available at Open Science Framework repository (OSF, https://osf.io/9drpj/).

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
