# Peer review of "In Silico and In Vitro Analyses Validate Human MicroRNAs Targeting the SARS-CoV-2 3′-UTR"

_ijms, 2021, doi:10.3390/ijms22116094_

Round 1

Reviewer 1 Report

The manuscript is addressing an important question about the interaction of 3’UTR of SARS-CoV-with human microRNAs.

Performing both in silico and in vitro analysis is one of the highlights of study.

However, the following comments are necessary to be addressed by Authors before publication:

1-

The following paper, recently published in Cells (MDPI) analyses the interaction of human microRNAs with 5’UTR of SARS-CoV-2.

 (2021). A Transcription Regulatory Sequence in the 5′ Untranslated Region of SARS-CoV-2 Is Vital for Virus Replication with an Altered Evolutionary Pattern against Human Inhibitory MicroRNAs. Cells, 10(2), 319.

The above paper needs to be cited and discussed. Particularly, is there any microRNA that can target both 5’UTR and 3’UTR?

2-The authors selected 60 SARS-CoV-2 sequences. However, there is a confusion here and needs extra explanation.

The authors have mentioned that:

“ 60 SARS-CoV-2 sequences covering major variants from 32 countries sampled between November 2019 and February 2021 (Table S3 of the supplementary material)”

However, how authors reached to the conclusion that the 60 sequences cover variants of 32 countries?

More analysis and descriptions of variants are required.

3-Which human genes are target of the shortlisted 3’-UTR-interacting microRNAs? Are those genes antiviral? A paragraph in discussion is required.

4-The work could benefit from microRNA expression analysis in samples originated from COVID-19 infected human/animal samples. This shortcoming needs to be mentioned in discussion.

Author Response

Response to Reviewer 1:

Thanks for the comments. We have answered to each suggestion in detail and included the manuscript will look like after the changes (in italics). We have marked the changes all changes to the manuscript in red type. Some of the sentences may not correspond to the final text, as we have also included the responses to Reviewer 2.

The manuscript is addressing an important question about the interaction of 3’UTR of SARS-CoV-with human microRNAs. Performing both in silico and in vitro analysis is one of the highlights of study. However, the following comments are necessary to be addressed by Authors before publication:

Q1- The following paper, recently published in Cells (MDPI) analyses the interaction of human microRNAs with 5’UTR of SARS-CoV-2. Authors (2021). A Transcription Regulatory Sequence in the 5′ Untranslated Region of SARS-CoV-2 Is Vital for Virus Replication with an Altered Evolutionary Pattern against Human Inhibitory MicroRNAs. Cells, 10(2), 319. The above paper needs to be cited and discussed. Particularly, is there any microRNA that can target both 5’UTR and 3’UTR?

Answer Q.1.: thanks for the suggestion, this an interesting article. As required the reviewer, we have cited and discussed it as shown below:

Introduction, paragraph 3: “Indeed, published and pre-printed in silico studies have identified putative binding sites in SARS-CoV-2 genome for human microRNAs [10-14, Mohammadi-Dehcheshmeh et al., 2021].”

Discussion, paragraph 1, sentence 2: “… Various authors have proposed that endogenous cellular microRNAs may help fight COVID-19 infection and have predicted putative binding sites for human microRNAs in the SARS-CoV-2 genome using in silico tools [10-14, Mohammadi-Dehcheshmeh et al., 2021]…

Discussion, end of Paragraph 7: …. Accordingly, 3 of the microRNAs targeting SARS-CoV-2 (all but hsa-miR-138-5p) preserve the same binding sites in SARS 3’-UTR whereas they are completely absent from the 3’-UTR of MERS according to IntaRNA (data not shown). Recently, Mohammadi-Dehcheshmeh and cols. (2021) have identified microRNAs which may target leading and transcription regulatory sequences at the 5′-UTR of SARS-CoV-2 and other coronaviruses. According to the authors, these microRNAs may reduce viral RNA replication and be key to lower viral load in the infected cells. The identified microRNAs includes only one human microRNA, namely hsa-miR-5004-3p, which interestingly, presents binding sites in the 3’-UTR according to RNA hybrid (mfe=-25.9) and STarMiR (Logit.Prob=0.749). Although we have not analysed this microRNAs because it did not fulfil all the criteria established for the in silico analyses, hsa-miR-5004-3p would merit further attention due to its potential targeting of both 5’- and 3’-UTRs.

Conclusions: “... This study is highly preliminary and further analyses – from reporter assays of additional candidates (e.g. the interesting hsa-miR-5004-3p, Mohammadi-Dehcheshmeh et al., 2021) to microRNA expression and effects in COVID-19 infected cells and animals– are needed to validate the therapeutic potential of this approach…”

Q.2-The authors selected 60 SARS-CoV-2 sequences. However, there is a confusion here and needs extra explanation. The authors have mentioned that: “ 60 SARS-CoV-2 sequences covering major variants from 32 countries sampled between November 2019 and February 2021 (Table S3 of the supplementary material)”. However, how authors reached to the conclusion that the 60 sequences cover variants of 32 countries? More analysis and descriptions of variants are required.

Answer Q.2.: We agree with the referee that the sentence in its present form may be confusing. Actually, we compared the 3’UTR from Wuhan sequence with that of 60 sequences covering 36 variants and all GISAID clades described in March 2021. These sequences were sampled between November 2019 and February 2021 in 32 countries. Obviously, not all variants were sampled in every month or country. Trying to avoid the confusion mentioned by the reviewer we have rephrased the sentence shown below. Concerning the requirement of additional analysis of variants, we have updated the analyses to comprise 3 additional sequences from the GISAID Pango Lineages of the Indian variant (Pango Lineage B.1.617), which appeared since the original submission of the manuscript and is spreading during the last months. Descriptions of the lineages are far beyond the scope of this article and may be accessed in different online resources such as Pango Lineages (https://cov-lineages.org/index.html) or GISAID (https://www.epicov.org/epi3/frontend#1b6656). All these changes are reflected in the results sections (paragraph 4) as follows:

“To explore whether the 3’-UTR sequence from the Wuhan reference sequence is conserved across variants we compared it to the 3’-UTR of 63 SARS-CoV-2 sequences covering all GISAID clades and 39 Pangolin Lineages (including the “British”, “South African”, “Brazilian”, “Indian” variants) sampled between November 2019 and May 2021 (Table S3 of the supplementary material). CLUSTALW alignment revealed that the 3’-UTR of all SARS-CoV-2 sequences analyzed here were 100% identical in all but 19 sequences. Changes in these 19 sequences comprised punctual mutations in one (11 sequences) or multiple nucleotides (2 sequences) or gaps up to 10 nucleotides (6 sequences). Gaps at the end of the sequence (by the poly-A tail) were present in the 3 sequences of the “South African variant” (Pango Lineage B.1.351) that were included in the analysis whereas punctual mutations appear in 2 of the 4 sequences of the “British variant” (Pango Lineage B.1.1.7), in the 2 samples of the “Brazilian variant” (Pango Lineage P.1), in the 2 of the 7 sequences of the “Californian variant” (Pango Lineages B.1.429 and B.1.427), and in the 3 samples of the “Indian variant” (Pango Lineage B.1.617, sublineages B.1.617.1 and B.1.617.2). An additional change involved a G to U or A mutation in the nucleotide 29742 that appeared in 4 sequences from 4 different countries sampled from January to September 2020 and which may alter the 3’-UTR secondary structure according to Hosseini & McLellan [14]. However, contrary to the potential spreading of the B.1.351, B.1.429 and B.1.427, B.1.1.7 and P.1 variants, nucleotide 29742 mutation does not seem to be spreading in recent times. Therefore, these comparisons indicate that microRNAs identified to target the 3’-UTR of Wuhan reference sequence can also target most SARS-CoV-2 variants although attention should be paid to the effects of the mutations of the British, South African, Brazilian, Californian, and Indian variants.”

Q3-Which human genes are target of the shortlisted 3’-UTR-interacting microRNAs? Are those genes antiviral? A paragraph in discussion is required.

Answer Q.3.: Following the suggestion of the reviewer we have identified the targets of the microRNAs that have been somehow validated according to miRTarBase (http://mirtarbase.cuhk.edu.cn/php/index.php). Then, we have compared the so-obtained list with the list of human genes/proteins that respond to the infection of SARS-CoV-2 from H2V database (http://www.datjar.com:40090/h2v/) and with the human genes categorized within the Gene Ontology category Viral Process (GO:0016032) defined as: “A multi-organism process in which a virus is a participant. The other participant is the host. Includes infection of a host cell, replication of the viral genome, and assembly of progeny virus particles. In some cases the viral genetic material may integrate into the host genome and only subsequently, under particular circumstances, 'complete' its life cycle.” These comparisons yield the following results:

H2V (human protein changes after COVID infection)

  • miR-128: ACIN1 (change in phosphorilation), COLGALT1 (interacts with viral nsp1 protein), AP2S1 (change in ubiquitination)
  • miR-138: none
  • miR-365: none
  • miR-3941: AKAP11 (change in phosphorilation), BEX4 (change in ubiquitination).

GO: Viral Process

  • miR-128: APS2S1, EPHA2
  • miR-138: DEK, EED, EZH2
  • miR-365: CAV2
  • miR-3941: none

These results are commented in the discussion as follows:

“Gene reporter assays indicate that hsa-miR-3941, hsa-miR-128-1-5p, hsa-miR-365b-5p and hsa-miR-138-5p target the viral 3’-UTR and, therefore, they may control viral pathogenicity. To our knowledge, besides bioinformatic predictions, nothing has been published relating any of these microRNAs and COVID-19. Indeed, very little has been published on the relationships between any microRNA and SARS-CoV-2 beyond the results from bioinformatics analyses. However, published studies on microRNA function and activity as well as on their validated targets (http://mirtarbase.cuhk.edu.cn/php/index.php, last accessed in May 2021; Huang HY, Lin YC, Li J, Huang KY, Shrestha S, Hong HC, Tang Y, Chen YG, Jin CN, Yu Y, Xu JT, Li YM, Cai XX, Zhou ZY, Chen XH, Pei YY, Hu L, Su JJ, Cui SD, Wang F, Xie YY, Ding SY, Luo MF, Chou CH, Chang NW, Chen KW, Cheng YH, Wan XH, Hsu WL, Lee TY, Wei FX, Huang HD. miRTarBase 2020: updates to the experimentally validated microRNA-target interaction database. Nucleic Acids Res. 2020, 48(D1):D148-D154. doi: 10.1093/nar/gkz896) can provide evidence supporting or dismissing their therapeutic potential. In this respect, there is evidence on the activity of hsa-miR-3941 in the respiratory system associated with a potential antiviral mechanism. Hsa-miR-3941 appears downregulated in children with acute pneumonia and in vitro analyses have revealed that it inhibits the phosphoinositol 3-kinase (PI3K)/serine-threonine kinase (AKT) signalling response to protect A549 human alveolar basal epithelial cells from injury [30]. According to Kindrachuk and cols. [31], the PI3K/AKT signaling response promotes the replication of MERS-CoV and other viruses, whereas their inhibition is capable of blocking viral replication in vitro, probably through a mechanism related to the Warburg effect [32]. According to miRTarBase, hsa-miR-3941 includes AKAP11 and BEX4 among its validated targets, the former reducing its phosporilation state and the latter its expression after SARS-CoV-2 infection in vitro (H2V database, http://www.datjar.com:40090/h2v/, last accessed in May 2021; Zhou N, Bao J, Ning Y. H2V: a database of human genes and proteins that respond to SARS-CoV-2, SARS-CoV, and MERS-CoV infection. BMC Bioinformatics. 2021 Jan 7;22(1):18. doi: 10.1186/s12859-020-03935-2). However, their relation to SARS-CoV-2 infection remains unclear. Therefore, the therapeutic upregulation of hsa-miR-3941 may combine the direct blockage of viral expression through 3’-UTR targeting with the inhibition of PI3K/AKT pathway to reduce viral infection.

Among the extensive bibliography on miR-138-5p, much concerns cancer processes and central nervous system functioning. However, a few references also relate it to COVID-19, in particular, bioinformatics predictions indicate that miR-138-5p may target SARS-CoV-2 genes [33,34] as well as IL6 and IL-8 cytokines, two major actors in the COVID-19-induced “cytokine storm” associated with poor outcomes [35]. In addition, reports by Pan and cols. [36,37] indicate that miR-138-5p targets herpes simplex virus mRNA and is key to promote viral latency and reduce morbidity and mortality in mice. Moreover, hsa-miR-138-5p has validated targets –DEK, EED, and EZH2– among genes included in GO term viral process (GO:0016032 from Gene Ontology) (http:// http://geneontology.org/, last accessed in May 2021; Gene Ontology Consortium. The Gene Ontology resource: enriching a GOld mine. Nucleic Acids Res. 2021, 49(D1):D325-D334. doi: 10.1093/nar/gkaa1113). The histone H3K27 methyltransferases EZH2 is particularly interesting since its inhibitors seem capable to suppress infection with DNA and RNA viruses according to Arbuckle and cols. (Arbuckle JH, Gardina PJ, Gordon DN, Hickman HD, Yewdell JW, Pierson TC, Myers TG, Kristie TM. Inhibitors of the Histone Methyltransferases EZH2/1 Induce a Potent Antiviral State and Suppress Infection by Diverse Viral Pathogens. mBio. 2017 Aug 15;8(4):e01141-17. doi: 10.1128/mBio.01141-17). As a whole, evidence suggests that hsa-miR-138-5p may be a good candidate for therapeutics, capable to regulate both the viral replication and the associated inflammatory response, as it does in herpes simplex virus.

Much less is known about miR-365b-5p and miR-128-1-5p. According to Pan and cols. [38], overexpression of miR-128-1-5p induces tight junction structural damage and increases cellular and paracellular permeability, resembling the early stage processes of COVID-19 infection [39] and dismissing the potential antiviral effects of this microRNA. Evidence compiled in miRTarBase indicates that miR-128-1-5p has 3 validated targets –AC1N1, AP2S1, COLGALT1– among the proteins changing after viral infection according to H2V database. One such change involves the interaction of SARS-CoV-2 nsp1 protein with COLGALT1 to potentially antagonise the innate immune response (Gordon DE, Hiatt J, et al. Comparative host-coronavirus protein interaction networks reveal pan-viral disease mechanisms. Science. 2020 Dec 4;370(6521):eabe9403. doi: 10.1126/science.abe9403. Epub 2020 Oct 15. PMID: 33060197; PMCID: PMC7808408). Another COVID-19 induced change involves the deubiquitination of AP2S1 (Stukalov, A., Girault, V., Grass, V. et al. Multilevel proteomics reveals host perturbations by SARS-CoV-2 and SARS-CoV. Nature (2021). https://doi.org/10.1038/s41586-021-03493-4), a subunit of AP2 adapter protein complex for clathrin also included within the Gene Ontology term Viral Process, which may contribute to the viral entrance. GO term Viral Process also includes EPHA2, another miR-128-1-5p validated target employed by different viruses to enter cells (Wang J, Zheng X, Peng Q, Zhang X, Qin Z. Eph receptors: the bridge linking host and virus. Cell Mol Life Sci. 2020 Jun;77(12):2355-2365. doi: 10.1007/s00018-019-03409-6. Epub 2019 Dec 31. PMID: 31893311; PMCID: PMC7275029), although direct evidence in SARS-CoV-2 are still lacking. Therefore, the available evidence about miR-128-1-5p is somehow contradictory and is difficult to evaluate its therapeutic potential without further analyses. Regarding miR-365b-5p, available information is even scarcer. According to the miRTarBase database of validated targets, it targets Cav-2, a component of lipid rafts where RNA virus such as Hepatitis c replicate (Shi ST, Lee KJ, Aizaki H, Hwang SB, Lai MM. HCV RNA replication occurs on a detergent-resistant membrane that cofractionates with caveolin-2. J Virol. 2003 Apr;77(7):4160-8. doi: 10.1128/jvi.77.7.4160-4168.2003. PMID: 12634374; PMCID: PMC150636) which may support some antiviral activity.

Q.4-The work could benefit from microRNA expression analysis in samples originated from COVID-19 infected human/animal samples. This shortcoming needs to be mentioned in discussion.

Answer Q.4.: We agree with the reviewer in the importance of this shortcoming. We have mentioned it in the conclusions section together with another shortcomings already identified in the previous version. The section now reads as follows:

The combined results from our bioinformatics and gene reporter analyses together with the available bibliography indicate that microRNAs, particularly hsa-miR-3941 and hsa-miR-138-5p, could be helpful to reduce viral pathogenicity of major current variants of SARS-CoV-2 or even SARS, in the case of hsa-miR-3941. This study is highly preliminary and further analyses – from reporter assays of additional candidates (e.g. the interesting hsa-miR-5004-3p) to microRNA expression and effects in COVID-19 infected cells and animals– are needed to validate the therapeutic potential of this approach...”

Reviewer 2 Report

The manuscript by Barreda-Manso et al. deals with the in silico analyses of human miRNAs targeting the SARS-CoV-2 3’ UTR. The authors used several MicroRNA target prediction databases to find potential hits which were further analyzed using a gene reporter assay.

The following points need to be addressed:

Major points:

  1. Abstract: line 17: one questions why has-miR-138-5p? – where does the addition results from?

Same for line 181: why did you include this central nervous specific miR-138? How did you get to the idea to include it?  

  1. Introduction: briefly mention properties/function of miRNAs (target region, seed region etc).
  2. lines 79 to 88: Predicted miRNAs were found with miRbase release 22. Why was the second analysis of the miRNAs done with miRbase release 21?
  3. Include a scheme of the reporter construct. Is the 3’ UTR connected to the Firefly or Renilla luciferase? Under what control is the control luciferase to normalize?
  4. In general: Scattering is quite big for most of the data points which makes the data not very convincing. Was the difference more within triplicates or within experiments? The authors should add an additional supplementary figure that shows all 9 individual data points (with data points from one experiment in one color and data points from other experiments in different color).
  5. ‘List of all microRNAs’ from Table S1 should be included in the main text and not as supplementary.
  6. Legend Figure 3b: mention that these are MTT assay data.

7. lines 244  and Table 2: in addition to this in silico data the authors should analyze mutant 3’ UTRs in their reporter assay (at least one for ‘marked in red’ in table 2 – mutations that involve NT binding, and one for  ‘marked in red bold’ – for binding in seed region.

  1. There is another paper by Alam T and Lipovich L dealing with human miRNAs in SARS-CoV2. This manuscript has not been mentioned – also: the miRNAs found in this paper have not been found in the previous publication. Please mention and discuss!
  2. Discussion - miRNA is discussed as therapeutic: please add how could miRNA be provided in a therapeutic setting.

Additional points:

  1. Title: targeting the SARS-CoV-2 3’ UTR (instead of on 3’ UTR)
  2. lines 10+11: rephrase the sentences since single-stranded is somewhat double
  3. line 13: function miRNAs : besides reduce protein expression – also degradation RNA?!
  4. line 18: Available information: should this be own data (if so: obtained data) or from literature?
  5. line 20: with complementary antiviral or tissue repairing effects – where are the data for tissue repairing effects?
  6. line 55ff: rephrase for better English: for some of which already antiviral effects have been shown against influenza etc..
  7. line 118: Wuhan insteadof Wuham
  8. line 215: to exclude instead of to discard

Author Response

Response to Reviwer 2:

Thanks for the comments. We have answered to each suggestion in detail and included the manuscript will look like after the changes (in italics). We have marked the changes all changes to the manuscript in red type. Some of the sentences may not correspond to the final text, as we have also included the responses to Reviewer 1.

Comments and Suggestions for Authors

The manuscript by Barreda-Manso et al. deals with the in silico analyses of human miRNAs targeting the SARS-CoV-2 3’ UTR. The authors used several MicroRNA target prediction databases to find potential hits which were further analyzed using a gene reporter assay.

The following points need to be addressed:

Major points:

Q.1. Abstract: line 17: one questions why has-miR-138-5p? – where does the addition results from? Same for line 181: why did you include this central nervous specific miR-138? How did you get to the idea to include it?

Answer Q.1.: as we mention in the discussion, this is a microRNA that we are analysing at the lab and which we included as a control. To our surprise, miR-138-5p has a fairly good activity in the reporter assay so we decided to continue its analysis. We have clarified it the results section (heading 2.3, paragraph 1) as follows:

“…We also included in the analyses the central nervous system-specific hsa-miR-138-5p –a human microRNA currently under study in our laboratory for an unrelated research– as a negative control that did not meet the selection criteria (without binding sites in the SARS-CoV-2 3’-UTR…”

Q.1bis. Introduction: briefly mention properties/function of miRNAs (target region, seed region etc).

Answer Q.1bis.: We agree that some readers may require further details on microRNA biology. However, to avoid increasing the length of the article with texts that are not essential to understand the study, we have opted to add a reference to the recent review (2015) by Mohr and Mott which provides a fairly detailed description of microRNA biology (function, targeting, biogenesis, …) in the introduction, paragraph 2:

MicroRNA-based therapies represent an alternative that remains to be fully explored. MicroRNAs are short non-coding RNA sequences that bind to target messenger RNAs (mRNAs), causing their cleavage or translational block to modulate a range of fundamental cellular processes (for an updated description of microRNA biology see Mohr and Mott, 2015).

Q.2. lines 79 to 88: Predicted miRNAs were found with miRbase release 22. Why was the second analysis of the miRNAs done with miRbase release 21?

Answer Q.2.: This is the result of a misunderstanding from us. Actually, the high confidence analyses also correspond to miRBase release 22. We have corrected this mistake throughout the text.

Q.3. Include a scheme of the reporter construct. Is the 3’ UTR connected to the Firefly or Renilla luciferase? Under what control is the control luciferase to normalize?

Answer Q.3.: We have included a reference to the Addgene website where to inspect the Luciferase/Renilla reporter scheme in the Material & Method section. The text is added in the Subcloning section as follows:

“… Then, we inserted the SARS-CoV-2 3′-UTR sequence into the pmirGLO Dual-Luciferase miRNA Target Expression Vector (Promega, a scheme on the reporter construct is available at: http://www.addgene.org/vector-database/8236/) between the SacI and SalI restriction sites using the FastDigest restriction enzymes (Thermo Scientific, USA)…

Q.4. In general: Scattering is quite big for most of the data points which makes the data not very convincing. Was the difference more within triplicates or within experiments? The authors should add an additional supplementary figure that shows all 9 individual data points (with data points from one experiment in one color and data points from other experiments in different color).

Answer Q.4.: We have added an additional supplementary figure (Figure S1) that shows all individual data points from each experiment and condition represented in figures 2, 3, and 4. This new figure shows the variability within experiments as well as between experiments and reveals that the latter is higher than the former.

Q.5. ‘List of all microRNAs’ from Table S1 should be included in the main text and not as supplementary.

Answer Q.5.: Although it would be easy to include the full list of microRNAs from Table S1 (identified microRNAs from full miRBase release 22) in the main text, we consider that in that case we would also have to include the full list of microRNAs from Table S2 (high confidence miRNAs), which is actually much larger (>700 microRNAs). We consider that it is better to keep the balance between the two in silico approaches in the main text and to refer to the supplementary tables for further information.

Q.6. Legend Figure 3b: mention that these are MTT assay data.

Answer Q.6.: We agree with reviewer and we have modified the figure 3b legend as follows:

 “(b) On the other hand, MTT assay data showed that empty pmiRGLO or…”

Q.7. Lines 244  and Table 2: in addition to this in silico data the authors should analyze mutant 3’ UTRs in their reporter assay (at least one for ‘marked in red’ in table 2 – mutations that involve NT binding, and one for  ‘marked in red bold’ – for binding in seed region.

Answer Q.7.: we agree with the reviewer that analysing the mutations that are currently active would be an improvement to evaluate the therapeutic potential of these microRNAs. However, these experiments are far beyond our capabilities. We are a neuroscience laboratory with experience in microRNA analysis which have used our resources (scarce) and a 675€ grant from a regional foundation to try to contribute to analyse and identify potential treatments for COVID-19. Therefore, we have neither the time nor the resources to carry out these analyses (that involves the cloning and testing of all the suggested sequences).

Q.8. There is another paper by Alam T and Lipovich L dealing with human miRNAs in SARS-CoV2. This manuscript has not been mentioned – also: the miRNAs found in this paper have not been found in the previous publication. Please mention and discuss!

Answer Q.8.: As required the reviewer, we have cited the paper in the following paragraphs:

Introduction, paragraph 3: “Indeed, published and pre-printed in silico studies have identified putative binding sites in SARS-CoV-2 genome for human microRNAs [10-14, Alam and Lipovich, 2021].”

Discussion, paragraph 1: “… Various authors have proposed that endogenous cellular microRNAs may help fight COVID-19 infection and have predicted putative binding sites for human microRNAs in the SARS-CoV-2 genome using in silico tools [10-14, Alam and Lipovich, 2021]…”

Discussion, paragraph 2: “… The list also includes hsa-miR-1307-3p, predicted to regulate SARS-CoV-2 expression due to its high binding affinity to viral genome [23-26, Alam & Lipovich, 2021], its regulation of H5N1 expression [27,28], and its conservations across species, including humans [29]”.

Discussion, paragraph 3: “Surprisingly, the list of microRNAs effectively targeting SARS-CoV-2 3’-UTR does not include hsa-miR-4775 and hsa-miR-1307-3p, both with binding sites according to various computational target prediction analyses [22-26,29, Alam & Lipovich, 2021].  Conversely,…”

We have tried to incorporate some discussion on this article but it is fairly complicate without also discussing all other articles that have make interesting microRNA predictions during this last year of pandemics. Therefore, for the shake of balance, we have restricted the discussion of this article to just a very short mention in the conclusions section when we mention that:

“…To our surprise, very little has been advanced in this respect judging from the published records despite the urgent need for COVID-19 treatments and the recent technological advances for RNA-therapeutics (e.g. RNA nanoformulations) already employed during the pandemic (i.e. Pfizer and Moderna vaccines) and which should allow a successful and safe deliver of microRNAs mimetics or even antagonists (as has been proposed for miR-122) to infected cells…”

Q.9: Discussion - miRNA is discussed as therapeutic: please add how could miRNA be provided in a therapeutic setting.

Answer Q.9.: Following the suggestion of the reviewer we have included a mention to microRNA delivery therapeutics for COVID-19 in the conclusions section as follows:

“The combined results from our bioinformatics and gene reporter analyses together with the available bibliography indicate that microRNAs, particularly has-miR-3941 and has-miR-138-5p, could be helpful to reduce viral pathogenicity of major current variants of SARS-CoV-2 or even SARS, in the case of has-miR-3941. This study is highly preliminary and further analyses– from reporter assays of additional candidates (e.g. has-miR-5004-3p, Mohammadi-Dehcheshmeh et al., 2021) to microRNA expression and effects in COVID-19 infected cells and animals– are needed to validate the therapeutic potential of this approach. To our surprise, very little has been advanced in this respect judging from the published records despite the urgent need for COVID-19 treatments and the recent technological advances for RNA-therapeutics (e.g. RNA nanoformulations, see review in Neeraj Chauhan, Meena Jaggi, Subhash C. Chauhan & Murali M. Yallapu (2021) COVID-19: fighting the invisible enemy with microRNAs, Expert Review of Anti-infective Therapy, 19:2, 137-145, DOI: 10.1080/14787210.2020.1812385) already employed during the pandemic (i.e. Pfizer and Moderna vaccines) and which should allow a successful and safe deliver of microRNAs mimetics or even antagonists (as has been proposed for miR-122) to infected cells. We hope that the present study will contribute to foster the exploration of this approach.”

Additional points:

  1. Title: targeting the SARS-CoV-2 3’ UTR (instead of on 3’ UTR)

Answer: We agree with reviewer and we have rephrased the title to:

 “In silico and in vitro analyses validate human microRNAs targeting the SARS-CoV-2 3’-UTR”

  1. lines 10+11: rephrase the sentences since single-stranded is somewhat double

Answer: Following the advice of the reviewer, we have rephrased the text to:

 “COVID-19 pandemic is caused by the betacoronavirus SARS-CoV-2. The genome of this virus is composed of a single strand of RNA with 5’ and 3’-UTR flanking…”

  1. line 13: function miRNAs : besides reduce protein expression – also degradation RNA?!

Answer: we agree with the reviewer in that microRNA may degrade mRNA to reduce protein expression. However, up to 6 different ways of regulating mRNA have been described for microRNAs besides mRNA degradation (see, for example, figure 2 of Y. Zhang, et al., Progress in microRNA delivery, J. Control. Release (2013), http://dx.doi.org/10.1016/ j.jconrel.2013.09.015). Therefore, to avoid extending too much a marginal part of the abstract, we consider a better option to keep the original sentence with a minimal change:

MicroRNAs are endogenous post-transcriptional regulators that target mRNA to modulate protein expression and mediate cellular functions”.

  1. line 18: Available information: should this be own data (if so: obtained data) or from literature?

Answer: in fact, the information is a combination of our results and the published data from other authors.

  1. line 20: with complementary antiviral or tissue repairing effects – where are the data for tissue repairing effects?

Answer: sorry, this is dragged from a previous version. We have modified the text to refer to the available evidence described in the discussion. The text now reads as follows:

“… combine effective targeting of SARS-CoV-2 genome with complementary antiviral or protective effects in the host cells that make …”

  1. line 55ff: rephrase for better English: for some of which already antiviral effects have been shown against influenza etc..

Asnwer: we agree so we have rephrased the sentence as suggested. The sentence now reads as:

…genome for human microRNAs [10-14], for some of which antiviral effects have been shown against influenza A,…”

  1. line 118: Wuhan instead of Wuham

Answer: Modified according to the suggestion

  1. line 215: to exclude instead of to discard

We agree with reviewer and we have rephrased the sentences to:

 “In order to exclude that reduction…